# Batch Renormalization: Towards Reducing Minibatch Dependence in Batch-Normalized Models

**Sergey Ioffe**
Google
`sioffe@google.com`

## Abstract

Batch Normalization is quite effective at accelerating and improving the training of deep models. However, its effectiveness diminishes when the training minibatches are small, or do not consist of independent samples. We hypothesize that this is due to the dependence of model layer inputs on all the examples in the minibatch, and different activations being produced between training and inference. We propose Batch Renormalization, a simple and effective extension to ensure that the training and inference models generate the same outputs that depend on individual examples rather than the entire minibatch. Models trained with Batch Renormalization perform substantially better than batchnorm when training with small or non-i.i.d. minibatches. At the same time, Batch Renormalization retains the benefits of batchnorm such as insensitivity to initialization and training efficiency.

## 1 Introduction

Batch Normalization ("batchnorm" [6]) has recently become a part of the standard toolkit for training deep networks. By normalizing activations, batch normalization helps stabilize the distributions of internal activations as the model trains. Batch normalization also makes it possible to use significantly higher learning rates, and reduces the sensitivity to initialization. These effects help accelerate the training, sometimes dramatically so. Batchnorm has been successfully used to enable state-of-the-art architectures such as residual networks [5].

Batchnorm works on minibatches in stochastic gradient training, and uses the mean and variance of the minibatch to normalize the activations. Specifically, consider a particular node in the deep network, producing a scalar value for each input example. Given a minibatch $\mathcal{B}$ of $m$ examples, consider the values of this node, $x_1 \ldots x_m$. Then batchnorm takes the form:

$$\widehat{x}_i \leftarrow \frac{x_i - \mu_{\mathcal{B}}}{\sigma_{\mathcal{B}}}$$

where $\mu_{\mathcal{B}}$ is the sample mean of $x_1 \ldots x_m$, and $\sigma_{\mathcal{B}}^2$ is the sample variance (in practice, a small $\epsilon$ is added to it for numerical stability). It is clear that the normalized activations corresponding to an input example will depend on the other examples in the minibatch. This is undesirable during inference, and therefore the mean and variance computed over all training data can be used instead. In practice, the model usually maintains moving averages of minibatch means and variances, and during inference uses those in place of the minibatch statistics.

While it appears to make sense to replace the minibatch statistics with whole-data ones during inference, this changes the activations in the network. In particular, this means that the upper layers (whose inputs are normalized using the minibatch) are trained on representations different from those computed in inference (when the inputs are normalized using the population statistics). When the minibatch size is large and its elements are i.i.d. samples from the training distribution, this

difference is small, and can in fact aid generalization. However, minibatch-wise normalization may have significant drawbacks:

For *small minibatches*, the estimates of the mean and variance become less accurate. These inaccuracies are compounded with depth, and reduce the quality of resulting models. Moreover, as each example is used to compute the variance used in its own normalization, the normalization operation is less well approximated by an affine transform, which is what is used in inference.

*Non-i.i.d. minibatches* can have a detrimental effect on models with batchnorm. For example, in a metric learning scenario (e.g. [4]), it is common to bias the minibatch sampling to include sets of examples that are known to be related. For instance, for a minibatch of size 32, we may randomly select 16 labels, then choose 2 examples for each of those labels. Without batchnorm, the loss computed for the minibatch decouples over the examples, and the intra-batch dependence introduced by our sampling mechanism may, at worst, increase the variance of the minibatch gradient. With batchnorm, however, the examples interact at every layer, which may cause the model to overfit to the specific distribution of minibatches and suffer when used on individual examples.

The dependence of the batch-normalized activations on the entire minibatch makes batchnorm powerful, but it is also the source of its drawbacks. Several approaches have been proposed to alleviate this. However, unlike batchnorm which can be easily applied to an existing model, these methods may require careful analysis of nonlinearities [1] and may change the class of functions representable by the model [2]. Weight normalization [11] presents an alternative, but does not offer guarantees about the activations and gradients when the model contains arbitrary nonlinearities, or contains layers without such normalization. Furthermore, weight normalization has been shown to benefit from mean-only batch normalization, which, like batchnorm, results in different outputs during training and inference. Another alternative [10] is to use a separate and fixed minibatch to compute the normalization parameters, but this makes the training more expensive, and does not guarantee that the activations outside the fixed minibatch are normalized.

In this paper we propose *Batch Renormalization*, a new extension to batchnorm. Our method ensures that the activations computed in the forward pass of the training step depend only on a single example and are identical to the activations computed in inference. This significantly improves the training on non-i.i.d. or small minibatches, compared to batchnorm, without incurring extra cost.

## 2  Prior Work: Batch Normalization

We are interested in stochastic gradient optimization of deep networks. The task is to minimize the loss, which decomposes over training examples:

$$\Theta = \arg\min_{\Theta} \frac{1}{N} \sum_{i=1}^{N} \ell_i(\Theta)$$

where $\ell_i$ is the loss incurred on the $i$th training example, and $\Theta$ is the vector of model weights. At each training step, a minibatch of $m$ examples is used to compute the gradient

$$\frac{1}{m} \frac{\partial \ell_i(\Theta)}{\partial \Theta}$$

which the optimizer uses to adjust $\Theta$.

Consider a particular node $x$ in a deep network. We observe that $x$ depends on all the model parameters that are used for its computation, and when those change, the distribution of $x$ also changes. Since $x$ itself affects the loss through all the layers above it, this change in distribution complicates the training of the layers above. This has been referred to as internal covariate shift. Batch Normalization [6] addresses it by considering the values of $x$ in a minibatch $\mathcal{B} = \{x_{1...m}\}$. It then

normalizes them as follows:

$$\mu_{\mathcal{B}} \leftarrow \frac{1}{m} \sum_{i=1}^{m} x_i$$

$$\sigma_{\mathcal{B}} \leftarrow \sqrt{\frac{1}{m} \sum_{i=1}^{m} (x_i - \mu_{\mathcal{B}})^2 + \epsilon}$$

$$\widehat{x}_i \leftarrow \frac{x_i - \mu_{\mathcal{B}}}{\sigma_{\mathcal{B}}}$$

$$y_i \leftarrow \gamma \widehat{x}_i + \beta \equiv \text{BN}(x_i)$$

Here $\gamma$ and $\beta$ are trainable parameters (learned using the same procedure, such as stochastic gradient descent, as all the other model weights), and $\epsilon$ is a small constant. Crucially, the computation of the sample mean $\mu_{\mathcal{B}}$ and sample standard deviation $\sigma_{\mathcal{B}}$ are part of the model architecture, are themselves functions of the model parameters, and as such participate in backpropagation. The backpropagation formulas for batchnorm are easy to derive by chain rule and are given in [6].

When applying batchnorm to a layer of activations x, the normalization takes place independently for each dimension (or, in the convolutional case, for each channel or feature map). When x is itself a result of applying a linear transform $W$ to the previous layer, batchnorm makes the model invariant to the scale of $W$ (ignoring the small $\epsilon$). This invariance makes it possible to not be picky about weight initialization, and to use larger learning rates.

Besides the reduction of internal covariate shift, an intuition for another effect of batchnorm can be obtained by considering the gradients with respect to different layers. Consider the normalized layer $\widehat{x}$, whose elements all have zero mean and unit variance. For a thought experiment, let us assume that the dimensions of $\widehat{x}$ are independent. Further, let us approximate the loss $\ell(\widehat{x})$ as its first-order Taylor expansion: $\ell \approx \ell_0 + g^T \widehat{x}$, where $g = \frac{\partial \ell}{\partial \widehat{x}}$. It then follows that $\text{Var}[\ell] \approx \|g\|^2$ in which the left-hand side does not depend on the layer we picked. This means that the norm of the gradient w.r.t. a normalized layer $\|\frac{\partial \ell}{\partial \widehat{x}}\|$ is approximately the same for different normalized layers. Therefore the gradients, as they flow through the network, do not explode nor vanish, thus facilitating the training. While the assumptions of independence and linearity do not hold in practice, the gradient flow is in fact significantly improved in batch-normalized models.

During inference, the standard practice is to normalize the activations using the moving averages $\mu$, $\sigma^2$ instead of minibatch mean $\mu_{\mathcal{B}}$ and variance $\sigma_{\mathcal{B}}^2$:

$$y_{\text{inference}} = \frac{x - \mu}{\sigma} \cdot \gamma + \beta$$

which depends only on a single input example rather than requiring a whole minibatch.

It is natural to ask whether we could simply use the moving averages $\mu$, $\sigma$ to perform the normalization during training, since this would remove the dependence of the normalized activations on the other example in the minibatch. This, however, has been observed to lead to the model blowing up. As argued in [6], such use of moving averages would cause the gradient optimization and the normalization to counteract each other. For example, the gradient step may increase a bias or scale the convolutional weights, in spite of the fact that the normalization would cancel the effect of these changes on the loss. This would result in unbounded growth of model parameters without actually improving the loss. It is thus crucial to use the minibatch moments, and to backpropagate through them.

## 3 Batch Renormalization

With batchnorm, the activities in the network differ between training and inference, since the normalization is done differently between the two models. Here, we aim to rectify this, while retaining the benefits of batchnorm.

Let us observe that if we have a minibatch and normalize a particular node $x$ using either the minibatch statistics or their moving averages, then the results of these two normalizations are related by an affine transform. Specifically, let $\mu$ be an estimate of the mean of $x$, and $\sigma$ be an estimate of its

**Input:** Values of $x$ over a training mini-batch $\mathcal{B} = \{x_{1\ldots m}\}$; parameters $\gamma$, $\beta$; current moving mean $\mu$ and standard deviation $\sigma$; moving average update rate $\alpha$; maximum allowed correction $r_{\max}$, $d_{\max}$.

**Output:** $\{y_i = \text{BatchRenorm}(x_i)\}$; updated $\mu$, $\sigma$.

$$\mu_{\mathcal{B}} \leftarrow \frac{1}{m} \sum_{i=1}^{m} x_i$$

$$\sigma_{\mathcal{B}} \leftarrow \sqrt{\epsilon + \frac{1}{m} \sum_{i=1}^{m} (x_i - \mu_{\mathcal{B}})^2}$$

$$r \leftarrow \texttt{stop\_gradient}\left( \text{clip}_{[1/r_{\max}, r_{\max}]}\left( \frac{\sigma_{\mathcal{B}}}{\sigma} \right) \right)$$

$$d \leftarrow \texttt{stop\_gradient}\left( \text{clip}_{[-d_{\max}, d_{\max}]}\left( \frac{\mu_{\mathcal{B}} - \mu}{\sigma} \right) \right)$$

$$\widehat{x}_i \leftarrow \frac{x_i - \mu_{\mathcal{B}}}{\sigma_{\mathcal{B}}} \cdot r + d$$

$$y_i \leftarrow \gamma \, \widehat{x}_i + \beta$$

$$\mu := \mu + \alpha(\mu_{\mathcal{B}} - \mu) \qquad \text{// Update moving averages}$$

$$\sigma := \sigma + \alpha(\sigma_{\mathcal{B}} - \sigma)$$

**Inference:** $\qquad y \leftarrow \gamma \cdot \dfrac{x - \mu}{\sigma} + \beta$

**Algorithm 1:** *Training (top) and inference (bottom) with Batch Renormalization, applied to activation $x$ over a mini-batch. During backpropagation, standard chain rule is used. The values marked with* `stop_gradient` *are treated as constant for a given training step, and the gradient is not propagated through them.*

standard deviation, computed perhaps as a moving average over the last several minibatches. Then, we have:

$$\frac{x_i - \mu}{\sigma} = \frac{x_i - \mu_{\mathcal{B}}}{\sigma_{\mathcal{B}}} \cdot r + d, \quad \text{where } r = \frac{\sigma_{\mathcal{B}}}{\sigma}, \ \ d = \frac{\mu_{\mathcal{B}} - \mu}{\sigma}$$

If $\sigma = \text{E}[\sigma_{\mathcal{B}}]$ and $\mu = \text{E}[\mu_{\mathcal{B}}]$, then $\text{E}[r] = 1$ and $\text{E}[d] = 0$ (the expectations are w.r.t. a minibatch $\mathcal{B}$). Batch Normalization, in fact, simply sets $r = 1$, $d = 0$.

We propose to retain $r$ and $d$, but treat them as constants for the purposes of gradient computation. In other words, we augment a network, which contains batch normalization layers, with a per-dimension affine transformation applied to the normalized activations. We treat the parameters $r$ and $d$ of this affine transform as fixed, even though they were computed from the minibatch itself. It is important to note that this transform is identity in expectation, as long as $\sigma = \text{E}[\sigma_{\mathcal{B}}]$ and $\mu = \text{E}[\mu_{\mathcal{B}}]$. We refer to batch normalization augmented with this affine transform as *Batch Renormalization*: the fixed (for the given minibatch) $r$ and $d$ correct for the fact that the minibatch statistics differ from the population ones. This allows the above layers to observe the "correct" activations – namely, the ones that would be generated by the inference model. We emphasize that, unlike the trainable parameters $\gamma, \beta$ of batchnorm, the corrections $r$ and $d$ are not trained by gradient descent, and vary across minibatches since they depend on the statistics of the current minibatch.

In practice, it is beneficial to train the model for a certain number of iterations with batchnorm alone, without the correction, then ramp up the amount of allowed correction. We do this by imposing bounds on $r$ and $d$, which initially constrain them to 1 and 0, respectively, and then are gradually relaxed.

Algorithm 1 presents Batch Renormalization. Unlike batchnorm, where the moving averages are computed during training but used only for inference, Batch Renorm does use $\mu$ and $\sigma$ during training to perform the correction. We use a fairly high rate of update $\alpha$ for these averages, to ensure that they benefit from averaging multiple batches but do not become stale relative to the model parameters. We explicitly update the exponentially-decayed moving averages $\mu$ and $\sigma$, and optimize the rest of the model using gradient optimization, with the gradients calculated via backpropagation:

$$\frac{\partial \ell}{\partial \widehat{x}_i} = \frac{\partial \ell}{\partial y_i} \cdot \gamma$$

$$\frac{\partial \ell}{\partial \sigma_{\mathcal{B}}} = \sum_{i=1}^{m} \frac{\partial \ell}{\partial \widehat{x}_i} \cdot (x_i - \mu_{\mathcal{B}}) \cdot \frac{-r}{\sigma_{\mathcal{B}}^2}$$

$$\frac{\partial \ell}{\partial \mu_{\mathcal{B}}} = \sum_{i=1}^{m} \frac{\partial \ell}{\partial \widehat{x}_i} \cdot \frac{-r}{\sigma_{\mathcal{B}}}$$

$$\frac{\partial \ell}{\partial x_i} = \frac{\partial \ell}{\partial \widehat{x}_i} \cdot \frac{r}{\sigma_{\mathcal{B}}} + \frac{\partial \ell}{\partial \sigma_{\mathcal{B}}} \cdot \frac{x_i - \mu_{\mathcal{B}}}{m \sigma_{\mathcal{B}}} + \frac{\partial \ell}{\partial \mu_{\mathcal{B}}} \cdot \frac{1}{m}$$

$$\frac{\partial \ell}{\partial \gamma} = \sum_{i=1}^{m} \frac{\partial \ell}{\partial y_i} \cdot \widehat{x}_i$$

$$\frac{\partial \ell}{\partial \beta} = \sum_{i=1}^{m} \frac{\partial \ell}{\partial y_i}$$

These gradient equations reveal another interpretation of Batch Renormalization. Because the loss $\ell$ is unaffected when all $x_i$ are shifted or scaled by the same amount, the functions $\ell(\{x_i + t\})$ and $\ell(\{x_i \cdot (1+t)\})$ are constant in $t$, and computing their derivatives at $t = 0$ gives $\sum_{i=1}^{m} \frac{\partial \ell}{\partial x_i} = 0$ and $\sum_{i=1}^{m} x_i \frac{\partial \ell}{\partial x_i} = 0$. Therefore, if we consider the $m$-dimensional vector $\{\frac{\partial \ell}{\partial x_i}\}$ (with one element per example in the minibatch), and further consider two vectors $\mathrm{p}_0 = (1, \ldots, 1)$ and $\mathrm{p}_1 = (x_1, \ldots, x_m)$, then $\{\frac{\partial \ell}{\partial x_i}\}$ lies in the null-space of $\mathrm{p}_0$ and $\mathrm{p}_1$. In fact, it is easy to see from the Batch Renorm backprop formulas that to compute the gradient $\{\frac{\partial \ell}{\partial x_i}\}$ from $\{\frac{\partial \ell}{\partial \widehat{x}_i}\}$, we need to first scale the latter by $r/\sigma_{\mathcal{B}}$, then project it onto the null-space of $\mathrm{p}_0$ and $\mathrm{p}_1$. For $r = \frac{\sigma_{\mathcal{B}}}{\sigma}$, this is equivalent to the backprop for the transformation $\frac{x-\mu}{\sigma}$, but combined with the null-space projection. In other words, Batch Renormalization allows us to normalize using moving averages $\mu$, $\sigma$ in training, and makes it work using the extra projection step in backprop.

Batch Renormalization shares many of the beneficial properties of batchnorm, such as insensitivity to initialization and ability to train efficiently with large learning rates. Unlike batchnorm, our method ensures that that all layers are trained on internal representations that will be actually used during inference.

## 4   Results

To evaluate Batch Renormalization, we applied it to the problem of image classification. Our baseline model is Inception v3 [13], trained on 1000 classes from ImageNet training set [9], and evaluated on the ImageNet validation data. In the baseline model, batchnorm was used after convolution and before the ReLU [8]. To apply Batch Renorm, we simply swapped it into the model in place of batchnorm. Both methods normalize each feature map over examples as well as over spatial locations. We fix the scale $\gamma = 1$, since it could be propagated through the ReLU and absorbed into the next layer.

The training used 50 synchronized workers [3]. Each worker processed a minibatch of 32 examples per training step. The gradients computed for all 50 minibatches were aggregated and then used by the RMSProp optimizer [14]. As is common practice, the inference model used exponentially-decayed moving averages of all model parameters, including the $\mu$ and $\sigma$ computed by both batchnorm and Batch Renorm.

For Batch Renorm, we used $r_{\max} = 1$, $d_{\max} = 0$ (i.e. simply batchnorm) for the first 5000 training steps, after which these were gradually relaxed to reach $r_{\max} = 3$ at 40k steps, and $d_{\max} = 5$ at 25k

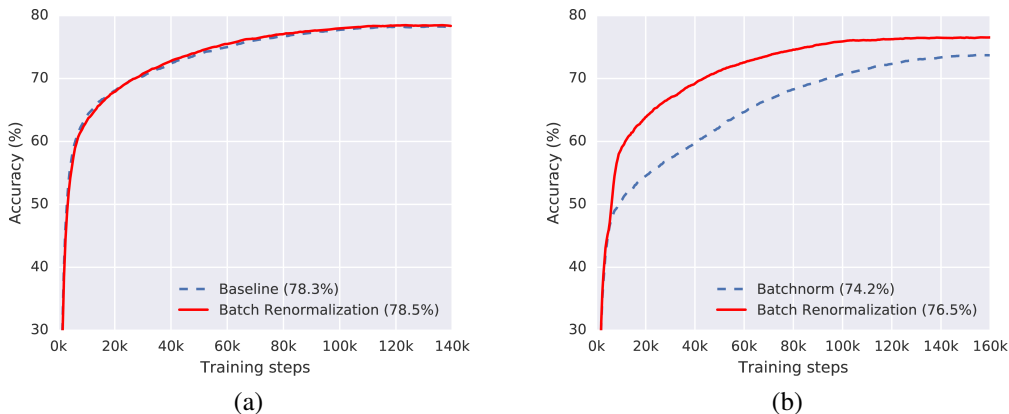

Figure 1: (a) *Validation top-1 accuracy of Inception-v3 model with batchnorm and its Batch Renorm version, trained on 50 synchronized workers, each processing minibatches of size 32. The Batch Renorm model achieves a marginally higher validation accuracy.* (b) *Validation accuracy for models trained with either batchnorm or Batch Renorm, where normalization is performed for sets of 4 examples (but with the gradients aggregated over all $50 \times 32$ examples processed by the 50 workers). Batch Renorm allows the model to train faster and achieve a higher accuracy, although normalizing sets of 32 examples performs better.*

steps. These final values resulted in clipping a small fraction of $r$s, and none of $d$s. However, at the beginning of training, when the learning rate was larger, it proved important to increase $r_{max}$ slowly: otherwise, occasional large gradients were observed to suddenly and severely increase the loss. To account for the fact that the means and variances change as the model trains, we used relatively fast updates to the moving statistics $\mu$ and $\sigma$, with $\alpha = 0.01$. Because of this and keeping $r_{max} = 1$ for a relatively large number of steps, we did not need to apply initialization bias correction [7].

All the hyperparameters other than those related to normalization were fixed between the models and across experiments.

## 4.1 Baseline

As a baseline, we trained the batchnorm model using the minibatch size of 32. More specifically, batchnorm was applied to each of the 50 minibatches; each example was normalized using 32 examples, but the resulting gradients were aggregated over 50 minibatches. This model achieved the top-1 validation accuracy of 78.3% after 130k training steps.

To verify that Batch Renorm does not diminish performance on such minibatches, we also trained the model with Batch Renorm, see Figure 1(a). The test accuracy of this model closely tracked the baseline, achieving a slightly higher test accuracy (78.5%) after the same number of steps.

## 4.2 Small minibatches

To investigate the effectiveness of Batch Renorm when training on small minibatches, we reduced the number of examples used for normalization to 4. Each minibatch of size 32 was thus broken into "microbatches" each having 4 examples; each microbatch was normalized independently, but the loss for each minibatch was computed as before. In other words, the gradient was still aggregated over 1600 examples per step, but the normalization involved groups of 4 examples rather than 32 as in the baseline. Figure 1(b) shows the results.

The validation accuracy of the batchnorm model is significantly lower than the baseline that normalized over minibatches of size 32, and training is slow, achieving 74.2% at 210k steps. We obtain a substantial improvement much faster (76.5% at 130k steps) by replacing batchnorm with Batch Renorm, However, the resulting test accuracy is still below what we get when applying either batchnorm or Batch Renorm to size 32 minibatches. Although Batch Renorm improves the training with small minibatches, it does not eliminate the benefit of having larger ones.

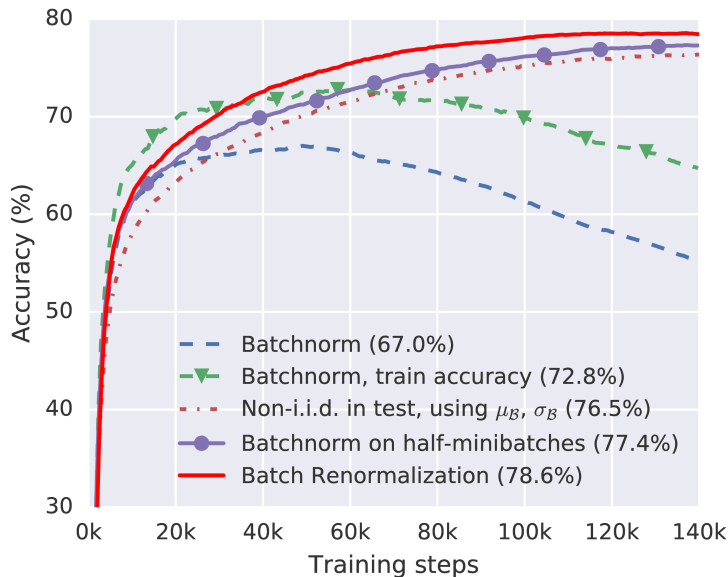

Figure 2: *Validation accuracy when training on non-i.i.d. minibatches, obtained by sampling 2 images for each of 16 (out of total 1000) random labels. This distribution bias results not only in a low test accuracy, but also low accuracy on the training set, with an eventual drop. This indicates overfitting to the particular minibatch distribution, which is confirmed by the improvement when the test minibatches also contain 2 images per label, and batchnorm uses minibatch statistics $\mu_{\mathcal{B}}$, $\sigma_{\mathcal{B}}$ during inference. It improves further if batchnorm is applied separately to 2 halves of a training minibatch, making each of them more i.i.d. Finally, by using Batch Renorm, we are able to just train and evaluate normally, and achieve the same validation accuracy as we get for i.i.d. minibatches in Fig. 1(a).*

### 4.3 Non-i.i.d. minibatches

When examples in a minibatch are not sampled independently, batchnorm can perform rather poorly. However, sampling with dependencies may be necessary for tasks such as for metric learning [4, 12]. We may want to ensure that images with the same label have more similar representations than otherwise, and to learn this we require that a reasonable number of same-label image pairs can be found within the same minibatch.

In this experiment (Figure 2), we selected each minibatch of size 32 by randomly sampling 16 labels (out of the total 1000) with replacement, then randomly selecting 2 images for each of those labels. When training with batchnorm, the test accuracy is much lower than for i.i.d. minibatches, achieving only 67%. Surprisingly, even the *training* accuracy is much lower (72.8%) than the *test* accuracy in the i.i.d. case, and in fact exhibits a drop that is consistent with overfitting. We suspect that this is in fact what happens: the model learns to predict labels for images that come in a set, where each image has a counterpart with the same label. This does not directly translate to classifying images individually, thus producing a drop in the accuracy computed on the training data. To verify this, we also evaluated the model in the "training mode", i.e. using minibatch statistics $\mu_{\mathcal{B}}$, $\sigma_{\mathcal{B}}$ instead of moving averages $\mu$, $\sigma$, where each test minibatch had size 50 and was obtained using the same procedure as the training minibatches – 25 labels, with 2 images per label. As expected, this does much better, achieving 76.5%, though still below the baseline accuracy. Of course, this evaluation scenario is usually infeasible, as we want the image representation to be a deterministic function of that image alone.

We can improve the accuracy for this problem by splitting each minibatch into two halves of size 16 each, so that for every pair of images belonging to the same class, one image is assigned to the first half-minibatch, and the other to the second. Each half is then more i.i.d., and this achieves a much better test accuracy (77.4% at 140k steps), but still below the baseline. This method is only

applicable when the number of examples per label is small (since this determines the number of microbatches that a minibatch needs to be split into).

With Batch Renorm, we simply trained the model with minibatch size of 32. The model achieved the same test accuracy (78.5% at 120k steps) as the equivalent model on i.i.d. minibatches, vs. 67% obtained with batchnorm. By replacing batchnorm with Batch Renorm, we ensured that the inference model can effectively classify individual images. This has completely eliminated the effect of overfitting the model to image sets with a biased label distribution.

## 5 Conclusions

We have demonstrated that Batch Normalization, while effective, is not well suited to small or non-i.i.d. training minibatches. We hypothesized that these drawbacks are due to the fact that the activations in the model, which are in turn used by other layers as inputs, are computed differently during training than during inference. We address this with Batch Renormalization, which replaces batchnorm and ensures that the outputs computed by the model are dependent only on the individual examples and not the entire minibatch, during both training and inference.

Batch Renormalization extends batchnorm with a per-dimension correction to ensure that the activations match between the training and inference networks. This correction is identity in expectation; its parameters are computed from the minibatch but are treated as constant by the optimizer. Unlike batchnorm, where the means and variances used during inference do not need to be computed until the training has completed, Batch Renormalization benefits from having these statistics directly participate in the training. Batch Renormalization is as easy to implement as batchnorm itself, runs at the same speed during both training and inference, and significantly improves training on small or non-i.i.d. minibatches. Our method does have extra hyperparameters: the update rate $\alpha$ for the moving averages, and the schedules for correction limits $d_{max}$, $r_{max}$. We have observed, however, that stable training can be achieved even without this clipping, by using a saturating nonlinearity such as $\min(\text{ReLU}(\cdot), 6)$, and simply turning on renormalization after an initial warm-up using batchnorm alone. A more extensive investigation of the effect of these parameters is a part of future work.

Batch Renormalization offers a promise of improving the performance of any model that would normally use batchnorm. This includes Residual Networks [5]. Another application is Generative Adversarial Networks [10], where the non-determinism introduced by batchnorm has been found to be an issue, and Batch Renorm may provide a solution.

Finally, Batch Renormalization may benefit applications where applying batch normalization has been difficult – such as recurrent networks. There, batchnorm would require each timestep to be normalized independently, but Batch Renormalization may make it possible to use the same running averages to normalize all timesteps, and then update those averages using all timesteps. This remains one of the areas that warrants further exploration.

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
