[Reviews · NeurIPS 2017]

Reviewer 1



In this paper, the authors propose Batch Renormalization technique to alleviate the problem of batchnorm when dealing with small or non-i.i.d minibatches. To reduce the dependence of large minibatch size is very important in many applications especially when training large neural network models with limited GPU memory. The proposed method is vey simple to understand and implement. And experiments show that Batch Renormalization performs well with non-i.i.d minibatches, and improves the results of small minibatches compared with batchnorm. Firstly, the authors give a clear review of batchnorm, and conclude that the key drawbacks of batchnorm are the inconsistency of mean and variance used in training and inference and the instability when dealing with small minibatches. Using moving averages to perform normalization would be the first thought, however this would lead to the model blowing up. So the authors propose a simple batch renormalization method to combine minibatch mean and variance with moving averages. In my opinion, what Batch Renormalization does is to gradually changing from origin batchnorm (normalizing with minibatch mean and variance) to batchnorm with (almost) only moving averages. In this way, the model can adopt part of the advantage of moving averages, and converge successfully. And I have several questions about this paper. (1) When using large minibatch size (such as 32), why Batch Renormalization has no advantage compared with batchnorm. It seems that the consistency of mean and variance in training and inference does not help much in this case. (2) Experiments show that the result of small minibatch size (batchsize=4) is worse that the result of large minibatch size (batchsize=32). So I wonder if using two (multiple) moving averages (mean and variance) with different update rates (such as one with 0.1, one with 0.01) would help. Small update rate helps to solve the inconstancy problem, large update rate helps to solve the small minibatch size problem. (3) The results of how r_max and d_max will affect the performace are not provided. There seems to have plenty of parameter tuning work. This work is good, and I am looking forward to seeing a more elegant solution to this problem.

Reviewer 2



This is an interesting paper, nice and neat. Batch normalization has proven to be an effective technique for dealing with internal covariate shift and has been widely used these days in the training of neural networks with a deep structure. It is known that BN has issues with small-size mini-batches. First of all, it gives rise to unreliable estimates of mean and variance of each mini-batch. Second, the mean and variance of the whole population, which is used in classification or inference, is computed by a moving average of mini-batches during training. This is a mismatch. This paper proposes a simple way to cope with the two issues in the conventional BN. It introduces another affine transform to correct the bias between the local and global normalization and this so-called renormalization makes the normalization in training and inference matched. The idea is simple but it seems to be fairly effective from the reported experimental results. I have no real criticism to lay out regarding the paper. However, I think the experiments can be more convincing if more results can be reported on mini-batches with a variety of sizes to give a full picture on the behavior of this batch renormalization technique. I am particularly curious about the case where the size of a mini-batch is down to one. In this case, the conventional batch normalization doesn't work any more but this renormalization can still be applied.

Reviewer 3



This work is an important contribution to improving SGD training of the Neural Networks. One remark I would like to make: Renormalizing moving-average affine transformation A(r,d) and the output affine transformation A(beta,gamma) form a composition which is also an affine transformation (see Algorithm 1, page 4). Therefore, one can "hide" (r,d) transformation inside the redefined (beta, gamma). Thus, the renormalization becomes a modification of the (beta,gamma) training algorithm.